# Time-reversal symmetry breaking type-II Weyl state in YbMnBi$_2$

Sergey Borisenko [1], Daniil Evtushinsky[1,2], Quinn Gibson[3,4], Alexander Yaresko [5], Klaus Koepernik[6], Timur Kim [7], Mazhar Ali [3], Jeroen van den Brink[6,8], Moritz Hoesch[7,9], Alexander Fedorov[1], Erik Haubold[1], Yevhen Kushnirenko [1], Ivan Soldatov [10,11], Rudolf Schäfer[10] & Robert J. Cava[3]

Spectroscopic detection of Dirac and Weyl fermions in real materials is vital for both, promising applications and fundamental bridge between high-energy and condensed-matter physics. While the presence of Dirac and noncentrosymmetric Weyl fermions is well established in many materials, the magnetic Weyl semimetals still escape direct experimental detection. In order to find a time-reversal symmetry breaking Weyl state we design two materials and present here experimental and theoretical evidence of realization of such a state in one of them, YbMnBi$_2$. We model the time-reversal symmetry breaking observed by magnetization and magneto-optical microscopy measurements by canted antiferromagnetism and find a number of Weyl points. Using angle-resolved photoemission, we directly observe two pairs of Weyl points connected by the Fermi arcs. Our results not only provide a fundamental link between the two areas of physics, but also demonstrate the practical way to design novel materials with exotic properties.

[1] Institute for Solid State Research, Leibniz IFW Dresden, Helmholtzstr. 20, 01069 Dresden, Germany. [2] Institute of Physics, Ecole Polytechnique Federale Lausanne, CH-1015 Lausanne, Switzerland. [3] Department of Chemistry, Princeton University, Princeton, NJ 08544, USA. [4] Department of Chemistry, University of Liverpool, Liverpool L69 7ZX, UK. [5] Max-Planck-Institute for Solid State Research, Heisenbergstrasse 1, 70569 Stuttgart, Germany. [6] Institute for Theoretical Solid State Physics, Leibniz IFW Dresden, Helmholtzstr. 20, 01069 Dresden, Germany. [7] Diamond Light Source, Harwell Campus, Didcot OX11 0DE, UK. [8] Institute for Solid State Physics, TU Dresden, 01062 Dresden, Germany. [9] Deutsches Elektronen-Synchrotron DESY, Photon Science, Hamburg 22607, Germany. [10] Institute for Metallic Materials, Leibniz IFW Dresden, Helmholtzstr. 20, 01069 Dresden, Germany. [11] Institute of Natural Sciences, Ural Federal University, 620002 Ekaterinburg, Russia. Correspondence and requests for materials should be addressed to S.B. (email: S.Borisenko@ifw-dresden.de)

D iscovery of Dirac fermions in graphene and topological insulators[1,2] sparked intense interest in finding their three-dimensional (3D) and non-degenerate analogs. First 3D-Dirac semimetals have been experimentally realized in $Cd_3As_2$ and $Na_3Bi$[3–5], but to observe Weyl fermions, the degeneracy of the electronic states should be lifted and 3D-Dirac point should effectively become split. In this case the Schrödinger equation will be similar to a two-component Weyl equation, and the electrons in this state will behave as Weyl fermions[6,7]. These requirements can be fulfilled either in noncentrosymmetric or in magnetic materials with strong spin–orbit interaction. Many compounds have been nominated to host this exotic state[8–27] and the evidence for Weyl fermions and Fermi arcs has been found in noncentrosymmetric crystals[9–14]. However, in spite of numerous theoretical predictions and intense experimental search[15–27], magnetic Weyl semimetals could not be spectroscopically identified.

One possible recipe would be to select materials that contain structural elements capable of generating the required electronic structure, focusing on magnetic systems with a center of inversion. The most simple and well-known generator of the Dirac-like massless relativistic dispersions is the 2D network of Bi atoms. These layers should be then separated by other blocks containing magnetic ions to break the time-reversal symmetry (TRS). To ensure direct spectroscopic observation of the Weyl state by angle-resolved photoemission (ARPES) one would choose to stay close to the antiferromagnetic (AFM) order to avoid strongly magnetized domains which would influence the photoelectrons. This role could be perfectly played by manganese atoms, which tend to order antiferromagnetically. Finally, in order to avoid a perfect AFM, which formally breaks TRS but does not lift the degeneracy, strongly magnetic ions of rare earth elements should be introduced to the lattice as well.

On the basis of these considerations we have grown single crystals of $EuMnBi_2$ (a previously reported compound[28]) and $YbMnBi_2$ (not a previously reported compound). Our ARPES, magnetization and magneto-optical microscopy measurements clearly demonstrate the presence of time-reversal symmetry breaking in one of the materials, $YbMnBi_2$, in excellent agreement with the calculations considering a canted antiferromagnetic order. Two kinds of fully relativistic calculations reveal a number of Weyl points and predict $YbMnBi_2$ to be the first canted antiferromagnet hosting a magnetic Weyl state of the second type[29]. We detect the Weyl points as well as an extra state (Fermi arc) of the Fermi surface both experimentally by ARPES and theoretically by semi-infinite-slab calculations. Our findings provide evidence for a TRS-breaking-induced type-II Weyl state that is present in $YbMnBi_2$.

## Results
**Characterization and band-structure calculations.** In Fig. 1 we present the crystal and electronic structures of both materials as well as some of their basic physical properties. $EuMnBi_2$ and $YbMnBi_2$ crystallize in I4/mmm and P4/nmm structures, respectively, which are very similar, but with an important difference in mutual arrangement of the basic structural units. $EuMnBi_2$ has a body-centered and $YbMnBi_2$ has a primitive tetragonal lattice. Because of this difference the square Bi net (green atoms, Bi2) contains a mirror plane in the former and a glide plane in the latter, and the corresponding symmetry of the perturbing potential drastically alters the pristine dispersions of the states originating from the square Bi nets. In addition, the magnetic moment of Eu ions is considerably larger than that of the Yb ions.

The resistivities and magnetoresistances are compared in Fig. 1d, e. Both compounds are metallic and have large,

non-classical magnetoresistances as seen for other $AMnBi_2$ compounds. In $EuMnBi_2$, a sharp feature in the magnetoresistance is seen due to a metamagnetic transition in the Eu sublattice, as reported in ref. [28]. For $EuMnBi_2$ the magnetism below 300 K is only due to the Eu moments as the Mn moments are locked antiferromagnetically. Additional ordering of Eu atoms seen in resistivity at ~20 K implies that the ground state of $EuMnBi_2$ is well defined and all magnetic moments, Mn and Eu, are antiferromagnetically ordered. There is no hysteresis in any of the measurements.

In the $YbMnBi_2$ compound we see no strong magnetism between 50 and 300 K, however, below 50 K new features in the magnetization develop. We show MH curves for $YbMnBi_2$ in Fig. 1f. The behavior above 20,000 Oe should be all bulk effects and the curves for 300, 200, 100, and 50 K only show a very weak either diamagnetic or weak paramagnetic response to the field above 20,000 Oe. This implies the spins are locked in possibly perfect AFM order, as in $EuMnBi_2$. However, as we get to low temperature, the magnetic response picks up and becomes significantly more anisotropic signaling some new spin degrees of freedom. This would be a disruption of perfect antiferromagnetism due to, e.g., canting and is an evidence for time-reversal symmetry breaking in $YbMnBi_2$.

We performed different types of band-structure calculations using the two methods: linear muffin-tin orbitals (LMTO) and full-potential local-orbitals (FPLO). First, we discuss the results of the former method. Corresponding Brillouin zones (BZ) are shown in the insets to Fig. 1a. In order to understand general aspects of the electronic structure of both materials, we first carried out calculations treating both Mn and rare earth originated states as quasi-core. When spin–orbit coupling (SOC) is neglected, this approach yielded the linear dispersions generated almost exclusively by Bi2 $p$ states, which cross near or at the Fermi level[30]. This is because the corresponding bandwidth is large and the band is centered at the Fermi level. The momentum locus of such symmetry protected Dirac crossings is indicated inside the BZ as a green loop for $YbMnBi_2$, a line and a point for $EuMnBi_2$. Exactly such elements are needed to construct a Weyl state and both compounds indeed demonstrate a number of Dirac dispersions using this simple approach (Supplementary Figs. 1–4). At all other points the crossings are not protected and the energy gap opens. Since otherwise the Bi networks are identical, it is also clear from Fig. 1a that staggered geometry of the Yb atoms with respect to Bi net protects more Dirac crossings in $YbMnBi_2$ (nodal loop) than the coincident geometry of Eu atoms in $EuMnBi_2$ (nodal line and point).

The next step is to include the Mn $d$ magnetism and spin–orbit coupling to the computational scheme. As a first approximation, we consider the ordering of Mn moments as perfectly antiferromagnetic (Fig. 1a) and neglect the contributions of the $4f$ states of rare earth ions which are still treated as non-spin-polarized quasi-core states. The results of the fully relativistic band-structure calculations are presented in Fig. 1b. Three bands contribute to the formation of the Fermi surface (FS), 2D cross-sections of which are shown in Fig. 1c. Green and blue dispersions correspond mostly to Mn states while the red one represents discussed above Bi $p$-states. As is seen, spin–orbit and exchange interactions make the Dirac electronic states "massive" in both materials, i.e., opens the gaps in most parts of the BZ. Again, there are important differences. While the gap is equally large (~0.2 eV) across the BZ in $EuMnBi_2$ creating isolated "lenses" of FS, the gap in $YbMnBi_2$ is smaller (down to ~10 meV along the ΓM line), adding to the "lenses" small electron-like pockets near the X-points. The distinct separation of these two portions of the FS signals the opening of the gap also in $YbMnBi_2$ and, in particular, breaking of the nodal loop (green contour in

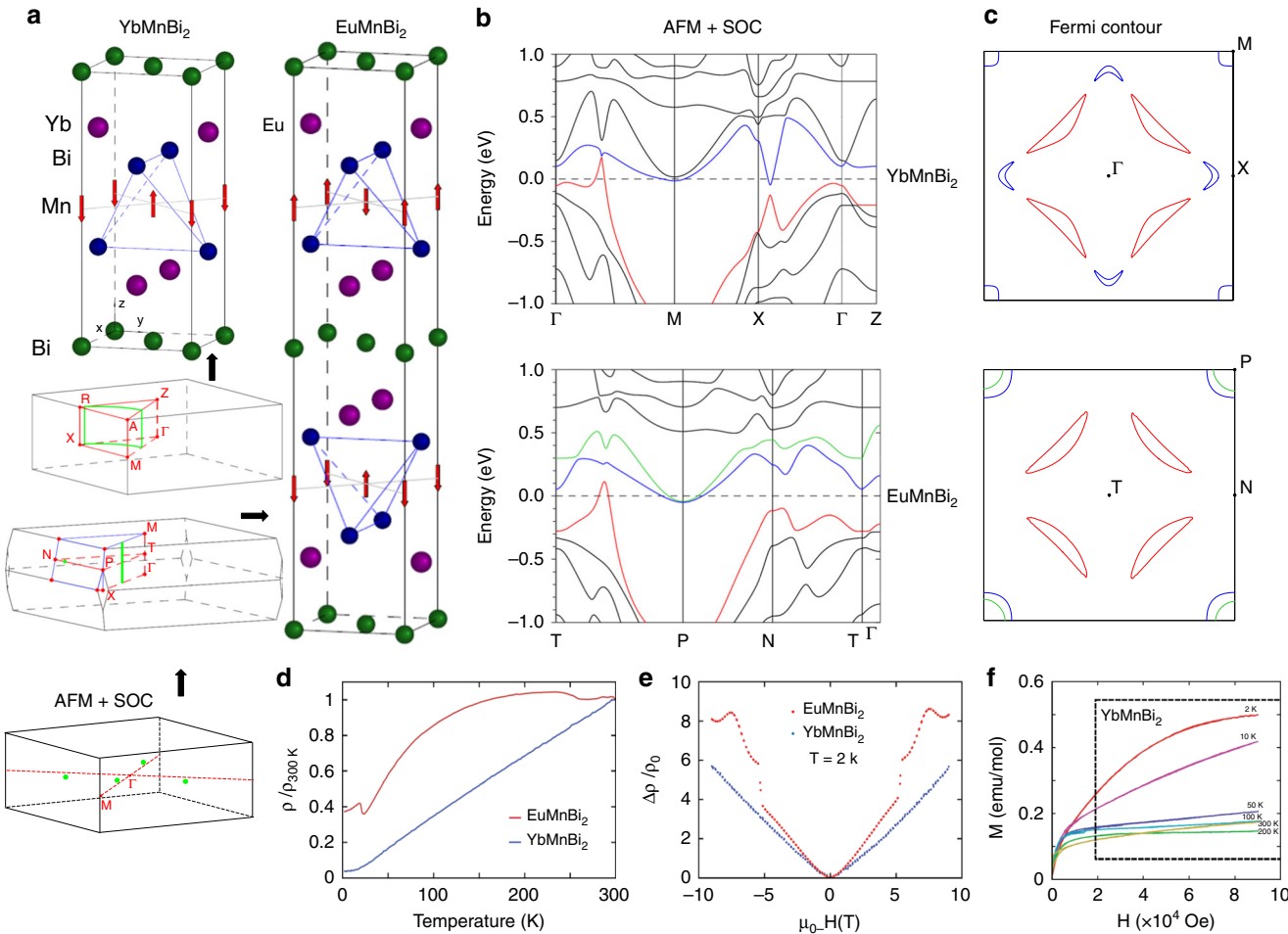

**Fig. 1** Calculated electronic structure and basic properties of EuMnBi$_2$ and YbMnBi$_2$. **a** Crystal structures of both materials together with the corresponding Brillouin Zones. Green lines and a point are a locus of Dirac crossings according to the calculations not taking into account spin–orbit interaction (Supplementary Figs 1–4). Red arrows are magnetic moments of Mn atoms. Inset shows the locations of 3D-Dirac points found in the AFM-SOC calculations. **b** Results of the fully relativistic band-structure calculations taking into account perfectly antiferromagnetic arrangement of Mn moments. **c** Corresponding Fermi contours for ΓXM and TNP high symmetry planes. **d** Resistivities and **e** Magnetoresistances of EuMnBi$_2$ and YbMnBi$_2$. **f** Magnetization curves showing the onset of the magnetism which breaks time-reversal symmetry in YbMnBi$_2$

upper BZ sketch in Fig. 1), but above E$_F$ one crossing is preserved as is seen from Fig. 1b where the red and blue dispersions touch each other along ΓM direction. We have found in total four 3D-Dirac points in YbMnBi$_2$ with the coordinates (±0.193, ±0.193, 0.015) and (±0.193, ±0.193, −0.015), which are schematically shown in lower BZ sketch in Fig. 1 as green points (Supplementary Fig. 5).

**Electronic structure.** In Fig. 2 we present the results of comparative study of Fermi surfaces and underlying dispersions by angle-resolved photoemission spectroscopy (ARPES). The FS of EuMnBi$_2$ (Fig. 2a) is remarkably similar to that resulted from the simplified calculations shown in Fig. 1c, apart from that Mn-states are absent at the Fermi level. This is not surprising since their computed energy localization is very sensitive to the value of the U parameter usually used to correctly reproduce the correlated nature of Mn 3d states. The well separated four lenses with sharp corners are clearly observed and depend only weakly on photon energy (Supplementary Fig. 6) confirming their origin from Bi p states from the two-dimensional square net. The ARPES intensity map taken along the cut #1 illustrates an ideally linear on the scale of 1 eV behavior of these features with

enormous Fermi velocity (9 eVÅ). Other states with considerably larger k$_z$ dispersions (which make them appear blurred because of intrinsically moderate k$_z$ resolution) are also seen, in accordance with the calculations. The cuts #2 and #3 emphasize another agreement with the theory—Dirac crossings are destroyed at all occupied momenta and sizeable energy gaps are opened in the vicinity of the Fermi level.

A qualitatively different picture is observed in the case of YbMnBi$_2$ (Fig. 2b). Although bearing a certain degree of similarity with the simplified calculations, the Fermi surface looks continuous and has a fine structure. The intensity distribution recorded along the cut #1 also shows extremely fast and sharp in momentum states originated from the Bi net, as in EuMnBi$_2$. Cut #2 runs through the additional FS elements and demonstrates the crucial dichotomy with the material containing Eu: the Dirac crossings appear to remain protected or the gap is much smaller. At particular photon energies (20.5, 27, 36, or 55 eV, Supplementary Note 1) the experimental FS contour of YbMnBi$_2$ looks sharp and continuous, i.e., hole-like "lenses" are connected to small electron-like pockets near X-points, implying the survival of the Dirac crossings in the momentum regions where these two FS sheets appear to be separated in Fig. 1c, at least for particular k$_z$. Here the crossings coincide with the Fermi

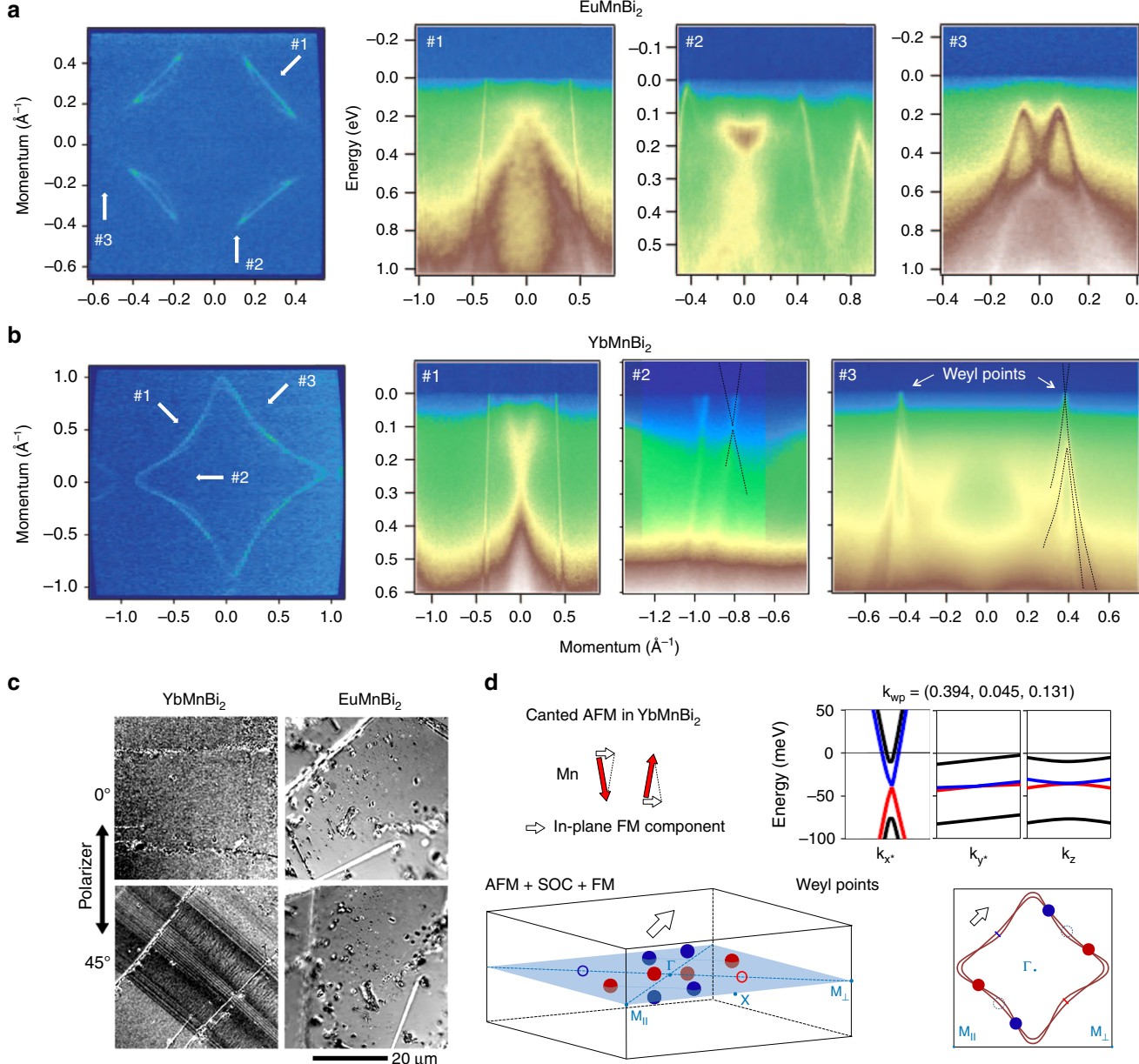

**Fig. 2** Experimental electronic structure of EuMnBi$_2$ and YbMnBi$_2$ and evidence for canting. **a** Fermi surface map of EuMnBi$_2$. White arrows with numbers indicate the directions along which the spectra in the panels to the right have been recorded. Data are taken using 27 eV photons (cut #1 at 44 eV). No Dirac crossings are seen in this case. **b** Same as (**a**), but for YbMnBi$_2$. Zero momentum in panel with the cut #2 ($h\nu = 80$ eV) corresponds to the BZ center. Dashed lines in the panel with cut #3 are guides to eye representing splitting of the degenerate bands due to TRS breaking. Crossings of these bands near the Fermi level are Weyl points. **c** Magneto-optical images of the YbMnBi$_2$ (left) and EuMnBi$_2$ (right) crystals, observed under conditions of the Voigt effect at perpendicular light incidence. A domain contrast can only be seen on the YbMnBi$_2$ material, proving the presence and the absence of magnetic moment canting in YbMnBi$_2$ and EuMnBi$_2$, respectively (see text). **d** Schematic results of the band-structure calculations taking into account the canting (10°) resulting in an in-plane ferromagnetic component along the (1,1,0) direction. $k_{x^*}$ and $k_{y^*}$ are the directions perpendicular to and along the Fermi surface contour, respectively

level and large gaps would be detectable. The continuous shape of the Fermi surface Fig. 2b is the consequence of the protection of the Dirac crossings at least in some points along the Fermi surface contour. By symmetry considerations, these Dirac crossings should not exist in the presence of SOC and TRS and inversion symmetries. Given no evidence of structural transitions, it is likely that TRS is broken in YbMnBi$_2$ also purely from the ARPES measurements of FS.

Obviously, it is instructive to take the data along the cut, which contains these special momentum regions. The next panel of Fig. 2b (cut #3) represents a crucial dataset, which underlines the

exotic electronic structure of YbMnBi$_2$ and directly proves the presence of the doubly (not quarterly) degenerate Dirac points, i.e., the long sought Weyl points. The first striking observation is that the Bi dispersions appear to be non-spin-degenerate in this particular region of the BZ. This breaking of the spin-degeneracy is key, and is not consistent with the AFM calculations. Indeed, the dispersion is seen to split between 100 and 200 meV binding energy and only one pair of split features is gapped while the other still reaches the Fermi level where they clearly cross. It is these crossing points sitting at the Fermi level are the essential components of the electronic structure of a Weyl

semimetal. Since these points are also the places where the electron-like FS touches the hole-like FS, YbMnBi2 is a canonical example of type-II Weyl semimetal[29]. We note that we managed to observe such a clear picture of crossing of non-degenerate bands only at particular photon energies (20, 27, and 31 eV, Supplementary Fig. 9).

The pronounced dichotomy between EuMnBi$_2$ and YbMnBi$_2$ in terms of degeneracy of the electronic states should be detectable by other experiments. We have studied both single crystals by means of magneto-optical microscopy. By employing a Kerr microscope[31], which is adjusted to be sensitive to the Voigt effect at perpendicular light incidence[32,33], it turned out to be possible to visualize areas ("domains") with different in-plane FM component of the Mn magnetization. This is demonstrated in the left column of Fig. 2c. If domains show up with maximum contrast at a given sample orientation (Fig. 2c), the contrast disappears upon sample rotation by 45°. This contrast symmetry is in agreement with the Voigt effect that is quadratic in the in-plane component of the magnetization (Supplementary Note 2). Since the measurements were carried out in zero magnetic field, it follows from Fig. 1f, that the averaging of the observed domains over the whole bulk sample will not result in significant net magnetization. Contrary to YbMnBi$_2$, no contrast can be observed at any sample orientation in case of the EuMnBi$_2$ crystal (Fig. 2c) as expected for a pure AFM in which the magnetic moments are aligned strictly perpendicular to the sample surface with no canting. The observed domain contrast in YbMnBi$_2$ cannot arise from structural features as the samples are single crystals and the cleavage planes of both materials are the same (Fig. 1a). The record sharpness of the quasi-2D ARPES features leaves no doubt as for their quality. The particular magnetic domain structure observed in the sample is most likely formed during the crystal growth process.

In order to understand the presence of the FM domains, magnetism seen in the MH curves and the origin of the observed lifting of the degeneracy by ARPES (TRS breaking), we have carried out calculations considering several types of canted antiferromagnetism of the Mn atoms—both with in-plane and out-of-plane ferromagnetic components. Remarkably, in all cases the degeneracy was lifted in the way it is observed experimentally: the gap between the linear Dirac dispersions became smaller for one of the non-degenerate components and disappeared in several momentum points. In other words, canting resulted in splitting of the 3D-Dirac points (Fig. 1a) into Weyl points and Weyl loops. In Fig. 2d we present the main results for the configuration of spins with the ferromagnetic component along ΓM direction (ΓM$_{II}$). Two sets of Weyl points are observed in this case. The first one consists of four points with the coordinates (0.193, 0.193, 0.12), (−0.193, −0.193, 0.12), (0.194, 0.194, −0.09), and (−0.194, −0.194, −0.09) with the energy of 168 meV above the Fermi level. These points move toward the mentioned earlier 3D-Dirac points upon decreasing of the canting angle and annihilate in the case of perfect antiferromagentism (Supplementary Table 2 and Fig. 2d). Another set of Weyl points with coordinates (0.394, 0.045, 0.131), (0.394, −0.045, 0.131), (0.045, 0.394, 0.131), and (−0.045, −0.394, 0.131) with the energy of 40 meV below the Fermi level corresponds to the crossing of the non-degenerate bands observed experimentally in Fig. 2b. In the right panels of Fig. 2d we show the band dispersions in the vicinity of one of these points. The key observation is that the Weyl "cones" are very anisotropic. If perpendicular to the Fermi surface in $k_x k_y$ plane, the Fermi velocity is a record-breaking one (9 eVÅ), if along the Fermi surface or along $k_z$, it is more than two orders of magnitude smaller (~0.043 eVÅ). Apart from the mentioned Weyl points we found the loops of Weyl crossings around the ΓM$_\perp$ (these are analogous to the Dirac loop shown

earlier in Fig. 1, but doubly degenerate, not quarterly, shown in Fig. 2d on the ΓM$_\perp$ line) and very small gap regions above the second set of Weyl points close to the ZAR plane. Our band-structure calculations thus clearly point to the first realization of the Weyl state in a canted antiferromagnet (Supplementary Note 3).

Taking into account the importance of the $k_z$-dispersion and very small gaps caused by anisotropic Weyl cones implied by the calculations, we have performed both very detailed photon energy dependent measurements and high-resolution measurements near the calculated Weyl points (Supplementary Note 1). It turned out that the mentioned earlier photon energies at which the FS contours were sharp and continuous approximately correspond to $k_z$-locations of either second set of Weyl points or the minimum gaps, naturally explaining the observed effect since the $k_z$-dispersion in these regions is negligible. However, even at $h\nu = 20$ eV we could not clearly resolve the tiny gaps and in both cases we observed qualitatively similar distribution of spectral weight near the Fermi level, as if the Weyl points existed at both $k_z$s.

**Fermi arcs**. As is shown in Fig. 2d, the Weyl points from the first set are projected to the surface BZ (dashed circles) such that they annihilate. The Weyl points from the second set, in contrast, should be connected by peculiar surface states (the Fermi arc), when measuring the 001 surface, as in our experiment. In order to detect the possible fine structure and elusive arc of surface states, we have used lower photon energies to increase momentum resolution. The high-precision Fermi surface map is shown in Fig. 3a. In the lower part of the map we overlay the experimental intensity with guides to eye to explain the observed features. There are clearly more features than one could previously conclude from Fig. 2b and simplified calculations presented in Fig. 1. While one can still identify the elements of the calculated FS—hole-like lenses and electron-like pockets close to the X-points, they appear connected and in the region of the "lens" one can clearly identify three features instead of two. If the connection between the lenses and electron-like pockets, as was shown earlier, is due to the lifting of the spin-degeneracy, the third feature on the map is the Fermi surface arc which connects two previously detected Weyl points. The data shown in Fig. 3b, c confirm this assignment. Indeed, there are three Dirac-like dispersions corresponding to each "lens". These two datasets are taken using different photon energy in an attempt to distinguish the bulk from the surface component. One can notice that in panel b two internal lines are less intense than the external ones, and in panel c the most inner ones are weaker than four others. On the basis of this observation and analysis of many other similar datasets we arrived at the conclusion that it is innermost linear dispersion corresponds to the Weyl arc. This is supported by the data taken from EuMnBi$_2$. We present enlarged fragments of panels b and c together with similar cuts taken from different sample in Fig. 3d. The third feature is clearly present in all datasets. In Fig. 3e the high-precision Fermi surface map is zoomed into the single lens. Not only is the arc remarkably absent, but also the shape and intensity distribution of the ARPES intensity helps us to identify the bulk component in YbMnBi$_2$. The bulk lens has a very strong straight section and corners. Exactly the same characteristics are peculiar to the outer dispersions in the discussed earlier panels (a) and (b) of Fig. 3. As expected, the energy-momentum cut in Fig. 3f contains only four dispersions, i.e., no signature of Weyl arc.

Figure 3a–c also suggests that two neighboring arcs are not equally resolved and their intensities are not exactly the same (Supplementary Fig. 14). This is expected, since the perpendicular

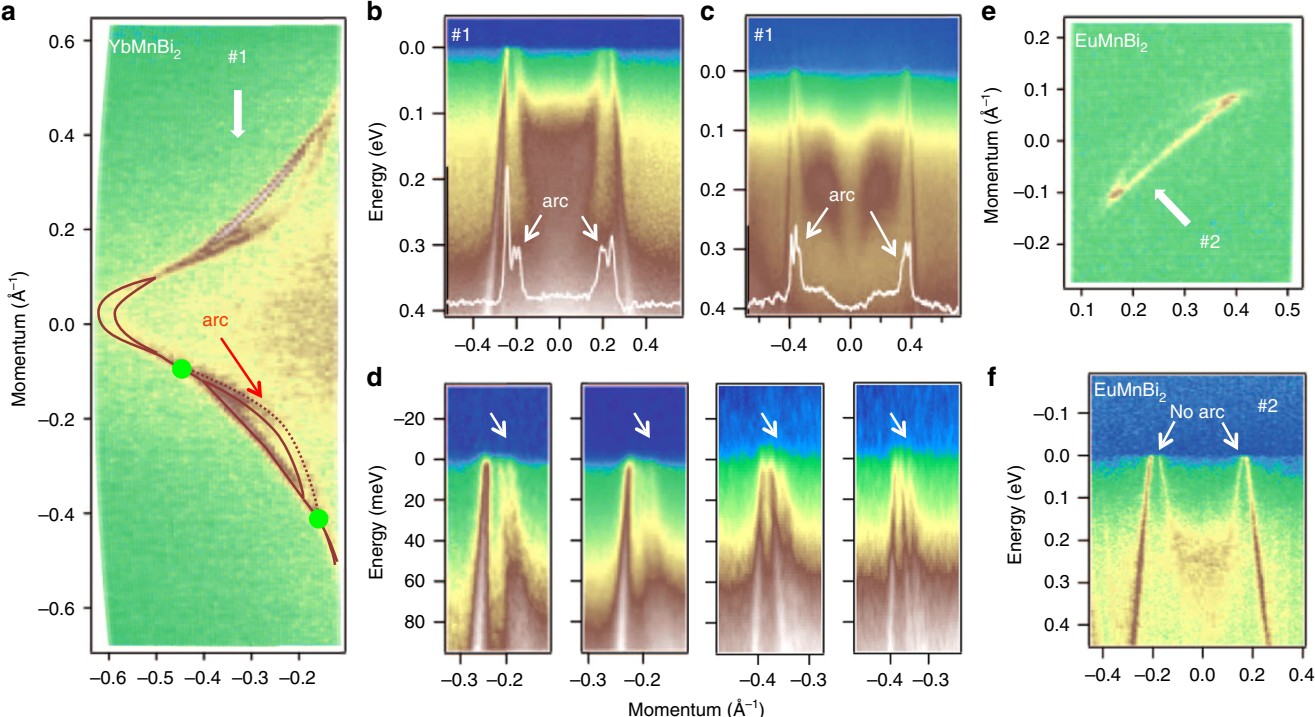

**Fig. 3** Fermi surface arc. **a** High-precision Fermi surface map of YbMnBi$_2$ at $h\nu = 20.5$ eV showing unusual number of features (brown lines) which can be explained by the presence of the arc (dotted line) corresponding to the surface states connecting the Weyl points (green points). **b** Spectra corresponding to cut #1 shown in panel (**a**) by white arrow ($h\nu = 20.5$ eV). White curve is the momentum-distribution curve at the Fermi level showing the presence of six crossings. The inner ones are due to the arc. **c** Same as in (**b**) but for $h\nu = 27$ eV. **d** Zoomed in fragments of panels (**b**) and (**c**) as well as similar data for other photon energies. **e** High-precision Fermi surface map of EuMnBi$_2$. **f** Spectra along the cut #2 indicated by white arrow in (**e**). Photon energy is 27 eV. No sign of the arc is present in the data

domains seen in Fig. 2c are not always equally strong and their contributions to the signal depend on the particular cleaved surface.

We briefly discuss other possible explanations of the unusual behavior seen in YbMnBi$_2$. We would like to immediately exclude the possible presence of slightly disoriented crystallites. First, the single crystals are of exceptional quality and at no stage during their characterization have we noticed the presence of such crystallites. Second, in this case one would observe doubling of all the features and we see only three. We also believe that the combined influence of finite $k_z$-resolution and $k_z$-dispersion cannot be responsible for the observed effect (the $z$-axis is perpendicular to the surface). The width of the linear dispersing features is resolution limited for all used photon energies (Supplementary Fig. 15) and $k_z$-dispersion related artifacts are usually much broader and not that clearly separated. In addition, the $k_z$-dispersion is very similar in Eu-containing material (see Fig. 1b) and there we do not see any evidence for additional features. Finally, we rule out the "technical" influence of the surface, such as relaxation[34], since, again, EuMnBi$_2$ cleaves in a similar way and there one would expect more surface-related problems because of much larger magnetic moments. We stress here that the decisive experimental fact, which speaks in favor of our interpretation, is the number of features at the Fermi level: any kind of artifact, including the relaxed surface, would produce a replica of the "lens" and thus double the number of crossings. We observe a "lens" and an arc, and the dispersion corresponding to the flat part of the former is a single feature at all photon energies (Supplementary Fig. 15). Previous ARPES study on closely related materials SrMnBi$_2$ and CaMnBi$_2$ isostructural to EuMnBi$_2$ and YbMnBi$_2$, respectively, did not reveal any surface states, but strongly anisotropic Dirac dispersions [35].

The connectivity and dispersion of the Fermi arc surface states depends sensitively on the surface orientation and termination[36]. The decisive supporting evidence for the existence of the arcs in YbMnBi$_2$ comes from the FPLO calculations. The results of these calculations taking into account spin–orbit coupling and antiferromagnetism of Mn atoms are fully consistent with previously discussed LMTO calculations with slightly different energy positions of the Mn-band and Dirac points (compare Fig. 1b with Supplementary Fig. 16). The corresponding Fermi surface is shown in Fig. 4a. Again, the lenses and electron-like pockets close to the X-points are well separated at all $k_z$'s highlighting the necessity of more sophisticated theoretical treatment to match ARPES data. Upon introducing the same canting angle of 10°, degeneracy is lifted and three symmetry inequivalent pairs of Weyl points appear (Fig. 4b). Unlike in the case of LMTO calculations, no small loops are present and a pair of very closely separated Weyl points (#3 and #4) is detected instead. The other two pairs are qualitatively the same. Importantly, the most separated in the **k**-space Weyl points (#5 and #6) are of the type II (Supplementary Fig. 17) and connect the non-degenerate lenses with electron pockets, exactly as in the experiment, although the calculated lenses are noticeably smaller (Fig. 4c). The Weyl points #1 and #2 are situated above each other and thus are not expected to contribute to the surface states on the measured 001 surface. The momentum separation of points #3 and #4 is too small to be resolved by our experiment.

The results of the finite-slab calculations of YbMnBi$_2$ presented in Fig. 4d, e prove the presence of the topological Fermi surface arcs, surface state dispersion parallel to bulk bands (as in Fig. 3) and clarify one important experimental observation. In a number of experiments on different samples we have always observed qualitatively similar to Fig. 3 picture as regards the Fermi arcs

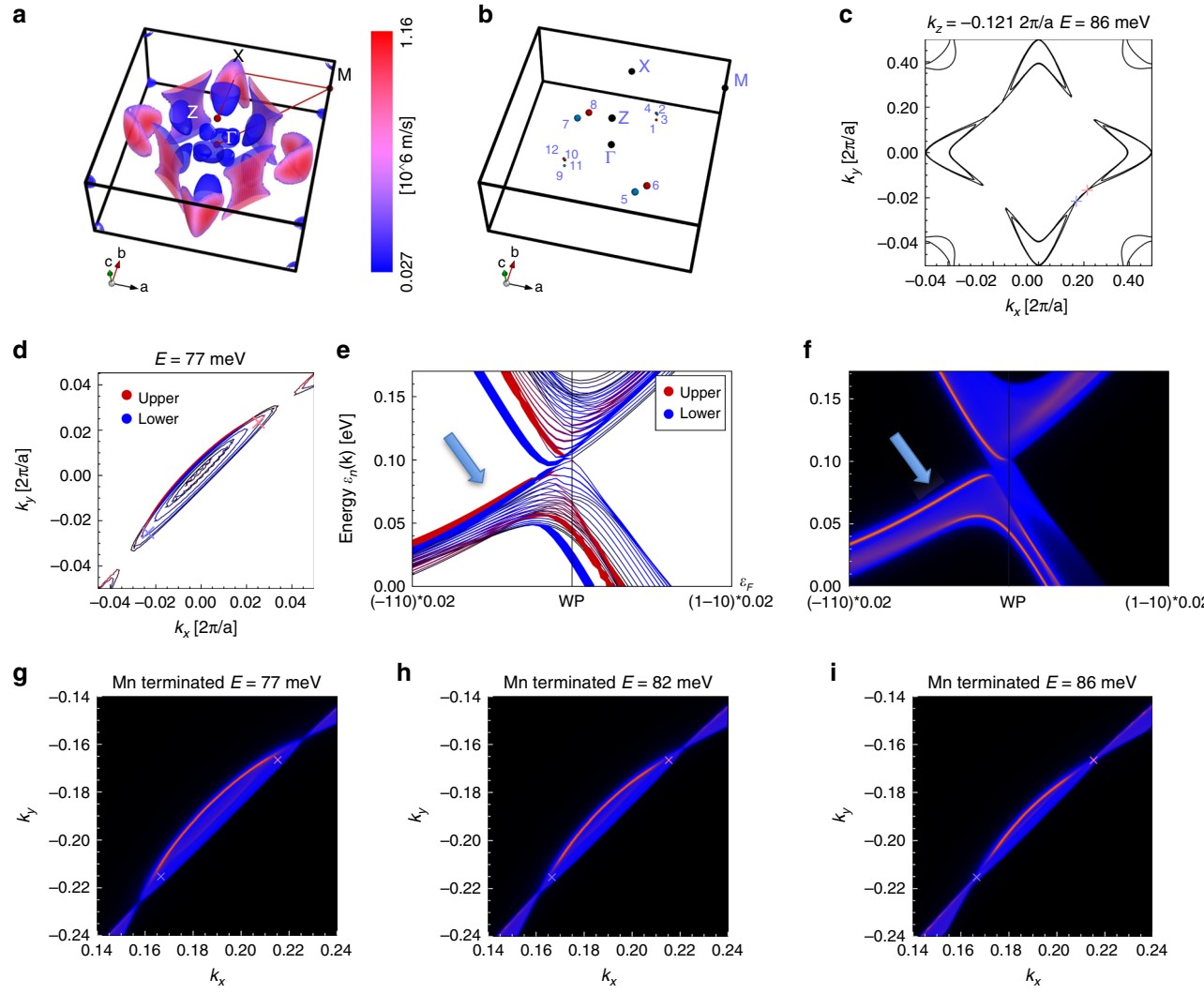

**Fig. 4** Calculated Fermi surface arcs (FPLO). **a** Three-dimensional representation of the Fermi surface of YbMnBi$_2$ without canting. Colors correspond to Fermi velocities. Blue Fermi surfaces in the middle of the BZ are highly 3D and are occasionally seen in different experimental FS maps as diffused intensity, e.g., Figure 3a and Supplementary Fig. 8. **b** Location of all Weyl points in the BZ for the case of 10° canting. The color of the WPs indicates the chirality. **c** FS contour for the $k_z$ corresponding to Weyl points #5 and #6, which are shown as small crosses in the places where the lens connects to the electron pocket. **d** FS contours of the bulk bands and arcs corresponding to the upper (red) and lower (blue) surfaces obtained in 14-unit-cell-slab calculations, at 77 meV (slightly below the WP energy). **e** Results of the same calculations as in (**d**) for the dispersion along a line perpendicular to the lens, cutting it at the midpoint between WPs #5 and #6. The arrow points to the closely separated topological surfaces states supporting the similar arcs on both surfaces. **f** Spectral function calculated along the same line as for (**e**) for a semi-infinite-slab in the limit of extreme surface sensitivity (12 $a_B$ penetration depth). A topological surface state is clearly visible as sharp red feature on the blue background of the bulk continuum. **g–i** Spectral function in momentum coordinates for different energies below and including the energy of the Weyl points #5 and #6. The Fermi arc is clearly seen as sharp red feature on the blue background of bulk FS crossings

location, implying that it does not strongly depend on which exactly surface is actually exposed. Figure 4d, e demonstrates that the arcs and supporting dispersions representing the upper (red) and lower (blue) surfaces are very close to each other and qualitatively similar.

We have also performed semi-infinite-slab calculations to model the spectral function seen in the surface-sensitive experiment. We show the results for the energy-momentum intensity distribution in Fig. 4f and momentum distribution for different energies in Fig. 4g–i. The maps at different energies are intended to illustrate the qualitative constancy of the picture upon increasing the size of the lens. The surface character of the well-defined arc and corresponding dispersion is clearly notable: narrow red features on the blue background of the $k_z$-integrated bulk bands. One should not expect a one-to-one correspondence

with our experimental data as far as the bulk states are concerned, since we have detected both rather narrow bulk features and their $k_z$-dispersion. This implies a finite $k_z$-resolution of our ARPES experiment and thus necessarily implies that the escape depth of photoelectrons is significantly larger than just one unit cell. Nevertheless, the presented results of the FPLO calculations directly support our ARPES data and thus the observation of the type II Weyl state in YbMnBi$_2$.

## Discussion

Now we will schematically illustrate the origin of the revealed Weyl state in YbMnBi$_2$ in Fig. 5 considering the gradual modification of the electronic and crystal structure of square net of Bi atoms. We start from the purely two-dimensional construct—the

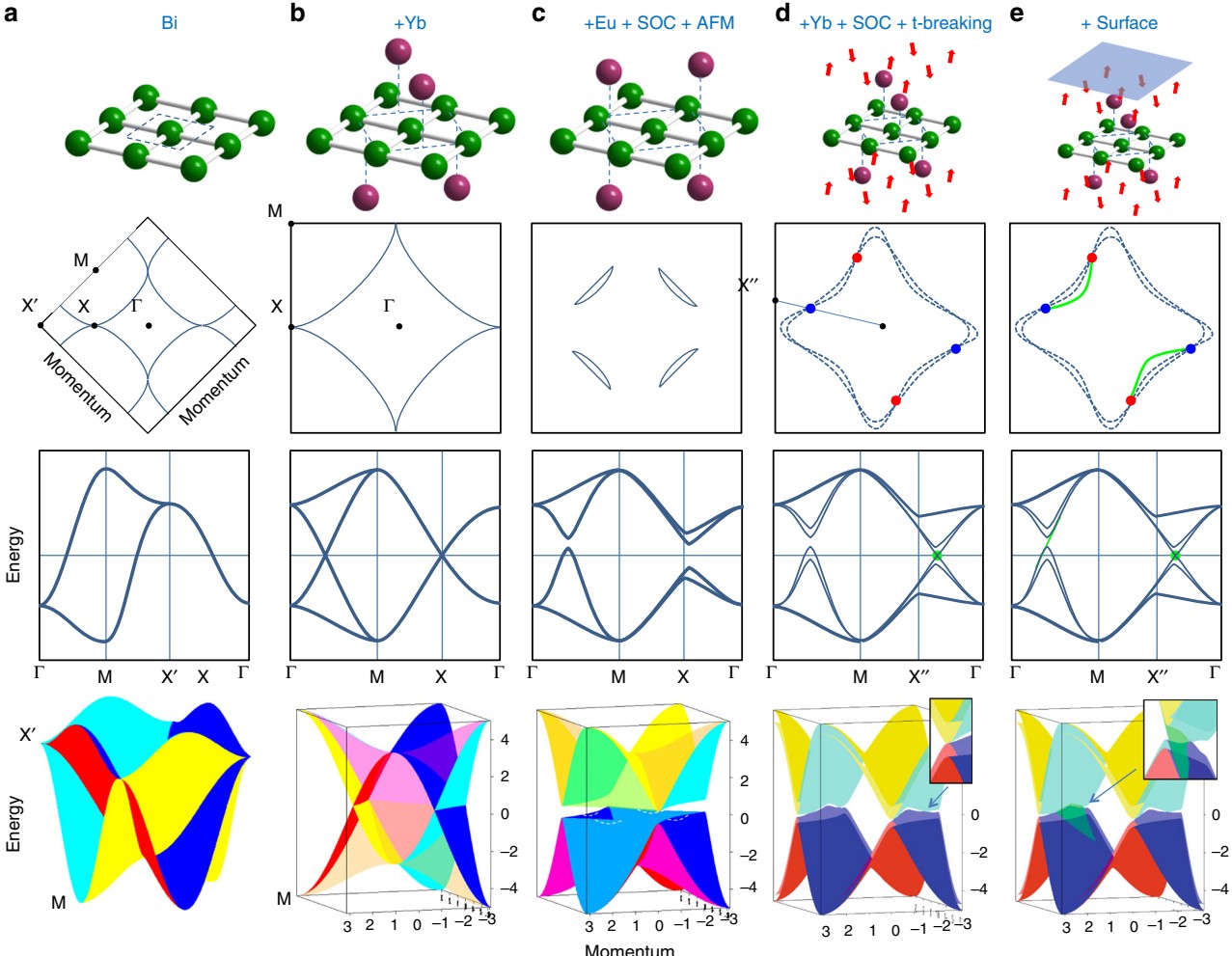

**Fig. 5** Emergence of the time-reversal type-II Weyl state. Schematic electronic structures corresponding to the structural fragment shown in the upper panel (from top to bottom): Fermi contour, band structure along selected directions, 3D E($\mathbf{k}$) representation in the whole BZ. **a** 2D network of Bi atoms. **b** Bi-network surrounded by Yb atoms in staggered geometry. **c** Bi-network surrounded by Eu atoms in coincident geometry. Spin–orbit coupling and perfect antiferromagnetism are taken into account in the calculations. White dashed lines in the lowest panel indicate where the sheets cross the Fermi level. **d** Case with Yb and canted AFM which results in lifted degeneracy and creation of Weyl points. Inset to the lowest panel shows the cut through one of the Weyl points. **e** Presence of the surface induces the non-trivial surface states (green planes) connecting the Weyl points (green points)

Bi net itself. Its electronic structure is shown in Fig. 5a. In this larger BZ there are no Dirac crossings and the dispersions are simple solutions of the tight-binding approach which takes into account only Bi $p$ states. Doubling of the unit cell because of the staggered coordination by Yb atoms results in Dirac crossings[30] exactly at the Fermi level and the momentum locus of these points is given by the diamond-shaped Fermi contour. This unique electronic structure is shown in three-dimensional space in the lowest panel of Fig. 5b and it is characterized by the 8-times-degenerate electronic states at X-point. It is remarkable that it is in the proximity of this point the degeneracy will be lifted completely in YbMnBi$_2$. In the next column of panels (Fig. 5c) we show the case of Eu material with SOC and perfect AFM inclu- ded. This example shows why Weyl state cannot be realized in EuMnBi$_2$: spin–orbit and exchange interactions open the energy gaps at all Dirac points. Here it is also easy to see the mechanism of the formation of the lenses: the gap is not constant and is not centered at the Fermi level. Next set of panels (Fig. 5d) represents the case of YbMnBi$_2$ as it is seen from ARPES. The value of $k_z$ corresponds to the plane containing Weyl points and the constant gap centered at the Fermi level is considered for simplicity. As mentioned earlier, the degeneracy is seen to be lifted leading to

formation of four pairs of Weyl points. Note that the cut ΓX"runs through the Weyl point. If the gap would be centered at the Fermi level along the diamond-like contour (Fig. 5b), YbMnBi$_2$ would have been a canonical Weyl semimetal of type I. Presence of lenses and electron pockets near X-points which touch in a single point make it the Weyl semimetal of type II. Finally, we present a schematic plot of the electronic states responsible for the Fermi arc together with the bulk-originated states in the panels of Fig. 5e. Again, for simplicity, we do not show all bands and the gap is chosen to be constant and centered at $E_F$. The arc at the Fermi surface is given by the surface states shown by green color. They originate from the bulk states forming the one Weyl point and terminate in bulk states in the vicinity of another Weyl point. The presence of four arcs in the sketch and experimental maps is due to the superposition of two domains.

Having demonstrated the realization of the Weyl state and Fermi surface arcs in YbMnBi$_2$ experimentally and theoretically, we would like to emphasize that further studies of this material are called for. It is not clear which precise magnetism results in time-reversal symmetry breaking, whether it is a bulk or only surface effect[37], and how to handle it theoretically. Our calcula- tions demonstrate that canted antiferromagnetism is one of the

possible solutions, but the exact configuration of spins is to be found. The chirality of Dirac electrons has already been discussed in related material with Sr instead of Yb (ref. [30]) and experimentally confirmed in YbMnBi$_2$ (ref. [38]). Due to extremely difficult ARPES experiments (low photoemission signal) and very high resolution needed to resolve the arcs, the use of spin-resolved modification of the technique is nearly blocked, meaning that direct observation of the chirality of the Weyl points is probably postponed until the further improvement of resolution of spin-resolved ARPES. However, the presence of the time-reversal symmetry breaking as seen by magnetization and magneto-optical microscopy, the existence of a continuous Fermi surface, a two-fold degenerate crossings at the Fermi level, and Fermi arc surface states that connect these crossings across the surface BZ are direct evidence of a TRS breaking induced Weyl state in YbMnBi$_2$ strongly supported by the band-structure and semi-infinite-slab calculations of the canted antiferromagnetic state; future work would be helpful in elucidating the exact mechanism of creating this state in this material.

## Methods

**Experiments.** High-quality single crystals were grown using a Bi rich melt in the ratios YbMnBi10 and EuMnBi10. The elements were heated to 1000 °C and cooled to 400 °C at 0.1 C/min, then subsequently centrifuged to remove excess Bi. The crystals exposed mirror-like portions of the surface after the cleave in ultra-high vacuum breaking z-periodicity. The structures of the compounds were solved by single crystal X-ray diffraction (Supplementary Table 3).

ARPES measurements were performed at the I05 beamline of Diamond Light Source, UK. Single crystal samples were cleaved in situ at a pressure lower than $2 \times 10^{-10}$ mbar and measured at temperatures about 7 K. Measurements were performed using (s,p)-polarized synchrotron light from 18 to 100 eV and employing Scienta R4000 hemispherical electron energy analyzer with an angular resolution of 0.2–0.5° and an energy resolution of 3–20 meV[39].

Magneto-optical imaging of magnetic domains at the surface of the YbMnBi$_2$ crystal was performed in a magneto-optical Kerr microscope adjusted for the Voigt effect at perpendicular light incidence at room temperature. White, linearly polarized light from a LED lamp was utilized.

**Calculations.** Band-structure calculations were performed for experimental crystal structures of YbMnBi$_2$ and EuMnBi$_2$ using the relativistic linear muffin-tin orbital method as implemented in the PY LMTO computer code[40]. Perdew-Wang[41] parameterization of the exchange-correlation potential in the local density approximation (LDA) was used. Localized 4f states of a rare earth ion were treated as quasi-core states assuming Yb2+(f14) and Eu2+(4f7) configurations.

AFM calculations were performed assuming that two Mn ions in the unit cell have opposite magnetization directions, i.e., without increasing the structural unit cell. For both compounds this results in checkerboard AFM order in ab plane. Magnetic order along c-axis is, however, FM for YbMnB$_2$ (P4/nmm) structure and AFM for EuMnB$_2$ (I4/mmm).

The FPLO band-structure calculations were obtained using the Full Potential Local Orbital code[42]. We treated the Yb 4f states as core states to remove their bands from the Fermi level. From the DFT results a minimum basis Wannier function model was extracted, which then was mapped onto the finite and semi-infinite slab geometries. The canting was introduced into the mapped model via auxiliary staggered magnetic fields at the Mn atoms in such a way that a 10° canting angle was obtained. The surface spectral functions were calculated via Greens function techniques. Following the procedures employed in ref. [43] we used the Berry curvature to prove the topological nature of all WPs.

## Data availability

All data are available from the corresponding author upon reasonable request.

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

## Acknowledgements

We are grateful to Denis Vyalikh, Geunsik Lee, Leonardo Degiorgi, Bernd Büchner, and Peter Armitage for the fruitful discussions, to Vladik Kataev and Stephan Zimmermann for performing ESR measurements, and to Kazuki Sumida, Tomoki Yoshikawa, and Taichi Okuda for performing spin-resolved ARPES measurements. This work was supported by DFG under the grant BO1912/7-1. The research at Princeton was supported by the ARO MURI on topological insulators, grant number W911NF-12-0461. We acknowledge Diamond Light Source for time on I05 under proposal SI11643-1.

## Author contributions

R.J.C., Q.G. and M.A. designed the candidate materials, single crystals, and results of characterization. S.B., D.E. and T.K. conceived the ARPES experiments. A.Y., K.K. and J.v.d.B. conducted theoretical studies. S.B., D.E., T.K., M.H., A.F., E.H. and Y.K. carried out ARPES measurements. I.S. and R.S. provided the results of magneto-optical microscopy measurements. S.B. and R.J.C. wrote the paper with the contributions from all the authors.

## Additional information

**Competing interests:** The authors declare no competing interests.

