## [Peer Review File · Nature Communications]

Editorial Note: This manuscript has been previously reviewed at another journal that is not operating a transparent peer review scheme. This document only contains reviewer comments and rebuttal letters for versions considered at Nature Communications .

Reviewers' comments:

Reviewer #3 (Remarks to the Author):

In the revised manuscript, the authors have addressed all of the points raised by the referee; (1) breaking of time reversal symmetry, and (2) three dimensionality of Weyl-cone band structure. For the point (1), the authors provide a solid experimental evidence for the weak ferromagnetism in YbMnBi_2 by showing the Kerr microscope images, in comparison with the perfect AFM case of EuMnBi_2 . For the point (2), the authors show a weak but clear k_z -dependence of the sharp band dispersion of Bi $6p$ orbital which creates Weyl points and a lens-type Fermi surface. Considering an importance of new spectroscopic evidence for the magnetic Weyl semimetal which has been hardly realized, I recommend publication of the manuscript to Nature Communications after additional revision as listed below.

- (1) The authors should detail the strength and direction of applied magnetic field in the Kerr microscope image in Fig. 2c. This clarifies the physical condition of the magnetic domain structure by referring to the M-H curve in Fig. 1f.
- (2) Figure 2c shows two directional stripe-type domains; 135-degree (from left-up to right down), and 45-degree (perpendicular to the former one) domains. While the ARPES data suggests that both domains are equally distributed at the cleaved surface, the 135-degree domain looks dominant in Fig. 2c. The authors should explain the reason for this inconsistency between the ARPES data and the Kerr microscope image.
- (3) A schematic of location of Weyl pairs shown in the bulk BZ with red and blue circles in Fig. 2d is confusing, particularly, it is hard to find which Weyl pair is aligned vertically so as not to contribute to the surface Fermi arc. The authors should improve this schematic.
- (4) Illustrations of Fermi surface in Fig. 5d and 5e are not accurate. The authors should change those to realistic ones with the C_2 -symmetry under a weak canting moment which breaks the C_4 symmetry of crystal.

Reviewer #3

We are grateful to the Referee for careful reading of the paper and useful recommendations to improve it.

(1) The authors should detail the strength and direction of applied magnetic field in the Kerr microscope image in Fig. 2c. This clarifies the physical condition of the magnetic domain structure by referring to the M-H curve in Fig. 1f.

We have added required information to the text describing Fig. 2c and made a connection to the M-H curve. The measurements were carried out in zero magnetic field to reproduce the experimental conditions. Fig. 1f thus implies that such domains are mostly compensating each other when averaged over the whole bulk sample.

(2) Figure 2c shows two directional stripe-type domains; 135-degree (from left-up to right down), and 45-degree (perpendicular to the former one) domains. While the ARPES data suggests that both domains are equally distributed at the cleaved surface, the 135-degree domain looks dominant in Fig. 2c. The authors should explain the reason for this inconsistency between the ARPES data and the Kerr microscope image.

Indeed, this is a very good point. It actually explains why the closest (no opposite) arcs in the BZ were always not equivalent. One can see this effect in Fig. 3 as well as in Fig. S12. That the difference is not that pronounced as in Fig. 2c is obviously connected with different cleaves. We added corresponding explanation to the text.

(3) A schematic of location of Weyl pairs shown in the bulk BZ with red and blue circles in Fig. 2d is confusing, particularly, it is hard to find which Weyl pair is aligned vertically so as not to contribute to the surface Fermi arc. The authors should improve this schematic.

Done.

(4) Illustrations of Fermi surface in Fig. 5d and 5e are not accurate. The authors should change those to realistic ones with the C2-symmetry under a weak canting moment which breaks the C4 symmetry of crystal.

Done.